

# Roof type classification with innovative machine learning approaches

Naim Ölçer, Didem Ölçer and Emre Sümer

Department of Computer Engineering, Başkent University, Ankara, Turkey

## ABSTRACT

Recently, convolutional neural network-based methods have been used extensively for roof type classification on images taken from space. The most important problem with classification processes using these methods is that it requires a large amount of training data. Usually, one or a few images are enough for a human to recognise an object. The one-shot learning approach, like the human brain, aims to effect learning about object categories with just one or a few training examples per class, rather than using huge amounts of data. In this study, roof-type classification was carried out with a few training examples using the one-time learning approach and the so-called Siamese neural network method. The images used for training were artificially produced due to the difficulty of finding roof data. A data set consisting of real roof images was used for the test. The test and training data set consisted of three different types: flat, gable and hip. Finally, a convolutional neural network-based model and a Siamese neural network model were trained with the same data set and the test results were compared with each other. When testing the Siamese neural network model, which was trained with artificially produced images, with real roof images, an average classification success of 66% was achieved.

## INTRODUCTION

Images taken from space or the air provide important information about the earth's surface. This information can be about buildings, land use areas and different land covers and is used in many fields, such as map creation, city planning, and climate change monitoring.

The roof is one of the most important parts of a building. Two of the most widely used methods of roof type classification in the literature are the convolutional neural network (CNN), which is a deep learning approach, and the support vector machine (SVM), a classical machine learning approach. However, a large amount of data is needed for training, especially in CNN-based methods. In this context, the one-shot learning (OSL) approach aims to accomplish the learning process with a small number of data samples, mimicking human learning.

In recent years, many roof-type classification studies have been carried out using CNN and SVM-based methods on aerial images. *Mohajeri et al. (2018)* used a traditional machine learning approach, the SVM method, to find a suitable space to place a solar panel. In this study, 10,085 pieces of roof data obtained from Geneva, Switzerland was used

Corresponding author
Naim Ölçer, naimolcer@gmail.com

as a data set and six roof types (flat & shed, gable, hip, gambrel & mansard, cross/corner gable & hip, and complex) were classified with an accuracy of 66%. In *Castagno & Atkins (2018)* study, CNN methods ResNet-50, Inception-ResNet-v2 and Inception-v3 were used. Roof data sets from Manhattan in New York City, Witten in Germany and Ann Arbor, Michigan were used in both RGB and LIDAR formats. The best result was achieved by using the ResNet-50 method on LIDAR data (88.3% accuracy). *Buyukdemircioglu, Can & Kocaman (2021)* aimed to first create a roof data set from very high-resolution (10 cm) orthophotos and then classify this data set using CNN architecture. In this study, a data set of approximately 10,000 images was used and six types of roofs (complex, flat, gable, half hip, hip, and pyramid) were classified. Results were obtained of 83%, 86%, 81%, 67%, 76% and 79% success rates, respectively. In another study, the results of the CNN and SVM methods were compared using images of the city of Munich from the WorldView-2 satellite for roof type estimation from high-resolution images (*Partovi et al., 2017*). As a result, 80% and 77% accuracy for the Flat roof type and 79% and 80% accuracy for the Gable roof type were obtained with CNN and SVM, respectively. Predicting the geometric properties of building roofs is a very important step because large-scale solar photovoltaic deployment on existing building roofs and neutral carbon emissions are one of the most efficient and viable renewable energy sources in urban areas (*Assouline, Mohajeri & Scartezzini, 2017*; *Lin et al., 2022*). *Assouline, Mohajeri & Scartezzini (2017)* proposed a multi-layered machine learning methodology to classify six roof types for all building roofs in Switzerland and they get an average accuracy of 67%. *Lin et al. (2022)* achieved 95.56% accuracy in building the rooftop recognition method with an improved DeepLAbv3+ network. In addition, recognition of the roof type is used to allow 3D modelling and rendering of the building (*Zang et al., 2015*; *Axelsson et al., 2018*). *Zang et al. (2015)* proposed an algorithm that splits a complex framework into parts and allows the system to recognize the entire framework based on the recognition of each part. They also trained their models using both synthetic and real roof images. Moreover, to solve problems such as detecting damage due to disasters or other external conditions, *Kim et al. (2021)* proposed a 43-layer CNN algorithm to detect roofs and classify materials. When their proposed CNN architecture was compared with a pretrained GoogleNet structure, they improved accuracy performance by 5–7%.

The concept of the Siamese neural network (SNN) first emerged in a study published in 1993 (*Bromley et al., 1993*). In this study, it was used on pairs of images for feature vector extraction. The similarity of the two input samples was also calculated. In another study, the SNN model and the OSL approach were used on the Omniglot data set, which holds alphabet data, and 92% accuracy was achieved (*Koch, Zemel & Salakhutdinov, 2015*). *Hsiao et al. (2019)* classified malware samples with the SNN model after converting them into images with pre-processing. This method has been found to be more successful than traditional methods. A CNN-based meta-learning framework was developed by *Zhang et al. (2021)* and the results of classifying the NWPU-RESISC45 and RSD46-WHU data sets were compared with those using dual CNNs. The developed CNN-based meta-learning framework achieved an accuracy of 69.46% with one shot and 84.66% with five shots. Meanwhile, *Chakrapani Gv et al. (2019)* studied the learning approach for

recognizing handwritten words and used the SNN model for model training. Using the George Washington data set and a data set consisting of Indian city names, a success rate of 92.4% was achieved with five shots. The basic one-shot hypothesis derivation (OSHD) approach developed by *Varghese et al. (2021)* was tested on two challenging computer vision tasks, Malayalam character recognition and diagnosis from retinal images, and the results were compared with those using the SNN. As a result, it has been observed that the proposed OSHD method gives better results than SNN. For building extraction, a multiscale Siamese network was used because of its usefulness for making predictions on unknown non-building class distributions even when few examples are available (*He et al., 2019*). *Wang et al. (2022)* proposed a study for one-shot retail product identification which is for the automatic recognition of retail products. For this purpose, they improved the Siamese network with a spatial channel dual attention. Their study showed that a method that can solve the data insufficient problem in the training stage on retail product identification outperforms traditional methods. For detecting changes in optical remote sensing images, *Yang et al. (2021)* proposed a supervised method based on a deep Siamese semantic segmentation network for 862 optical image pairs. Their experimental results showed that the proposed Siamese network is better than other methods for change detection problems.

In the literature, many studies predicting roof types with CNN models, need more data. However, due to limited data and region dependence, the motivation of this study is to explore the success of the OSL method by training smaller datasets in predicting roof types. In this study that forms the subject of this article, for the first time, building roof type classification *via* satellite images was undertaken according to the OSL approach. In another innovative aspect, sample artificial images of flat, gable and hip roof types were produced in the virtual environment, thereby eliminating the necessity of obtaining the training data required for the roof type classification from real images. Then, learning from a small number of image samples was carried out by using the SNN model and the trained model was tested. For the test process, real roof images taken from the satellite were used. The same data was used with the CNN-based DenseNet architecture, and performance metrics were compared.

## DATA SET

The world's buildings display a huge number of roof types, which vary depending on climatic conditions, building use type, regional architecture and aesthetics. Some of the most common roof types are flat, gable, hip, arch, and dormer.

In this study, Flat, Gable and Hip, which are the most fundamental roof types, were selected for classification. Some basic features of these selected roof types are given below. In addition, an example of each roof type is shown in Fig. 1.

**Flat**: A flat roof, unlike many pitched roof types, is nearly flat. They are straight, usually slightly curved, and found in traditional structures in regions with less rainfall.

**Gable:** A gable roof is a triangular roof consisting of two parts whose upper horizontal edges join to form a ridge. It is the most common form of roof used in cold or temperate climates.
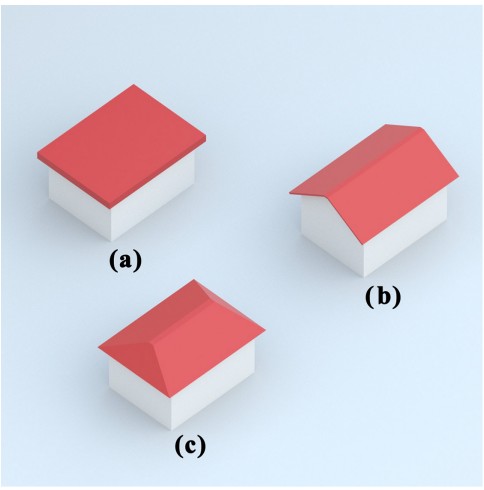

**Figure 1 Roof types—(A) flat (B) gable (C) hip.**

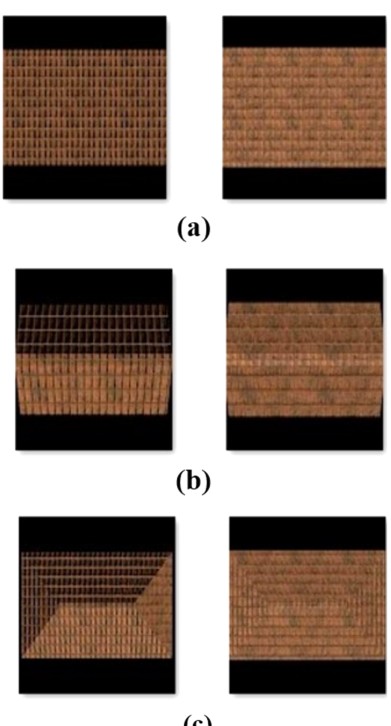

**Figure 2 Examples of artificially generated data set (A) flat (B) gable (C) hip.**

**Hip:** A hipped roof usually has a slight slope, with all the sides sloping downwards towards the walls. It is one of the common roof shapes in rainy, snowy, and very windy climates.

Artificially produced roof images such as those shown in Fig. 2 were used for model training due to the difficulty of finding a roof data set and the desire to make the study independent of region. These images are of three types, flat, gable, and hip, and were

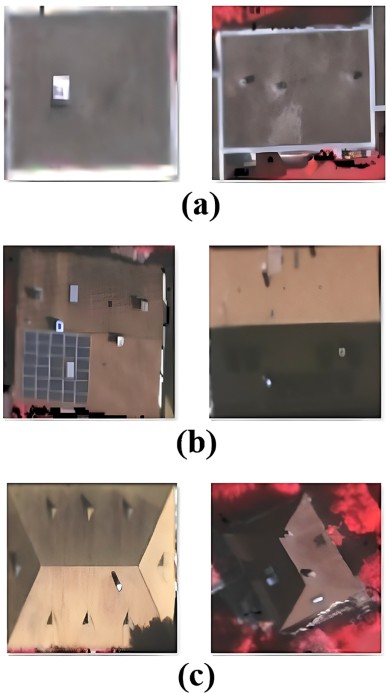

**Figure 3** **Examples of roof types from the test data set—(A) flat (B) gable (C) hip.**

produced by the authors using Autodesk Maya software (*Murdock, 2022*). The dataset can be accessed *via* https://github.com/olcernaim/roof-type-classification. To increase proximity to reality, images with three different sun angles and textures were produced for each roof type. In the test process, real roof images were used.

The same three roof types were included in the data set used for the test. The roof sub-types of Hip and Hip & Valley were in the Hip data set; the Gable data set included Gable, Gable with Dormer, and Gable & Valley. The test data set used in the study was taken from https://github.com/loosgagnet/Building-detection-and-roof-type-recognition (*Alidoost & Arefi, 2018*). Figure 3 shows two examples of each roof type from this data set.

During the data pre-processing step, the .tif images in the data set which were to be used were converted from 224 × 224 to 105 × 105, the size that the SNN model can use. They were then converted to .jpg format.

## METHODOLOGY

### Convolutional neural network

As a result of the increasing interest in deep learning in recent years, the use of CNN has become very widespread. CNN, which is frequently preferred especially because it can process large amounts of data, gives very successful results, especially to machine learning problems.

Image classification, object detection and object segmentation are examples of the main usage areas of CNN (*Krizhevsky, Sutskever & Hinton, 2017*; *Phung & Rhee, 2018*; *Ghosh et al., 2020*; *Simonyan & Zisserman, 2014*; *Szegedy et al., 2015*; *Zeiler & Fergus, 2014*; *He*

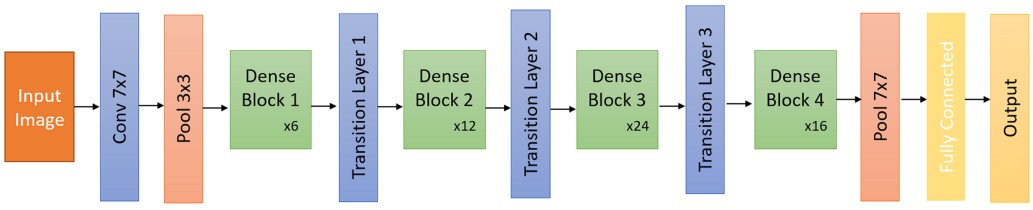

**Figure 4 DenseNet architecture.**

*et al., 2016*). In this study, the DenseNet architecture of CNN was chosen to be compared with the OSL approach. The main feature of DenseNet is that it is frequently used for visual object recognition problems. Another reason for choosing it is that it combines the output of the previous layer with the next layer by using dense blocks rather than the deep paths between input and output favoured by other deep learning architectures (*Huang et al., 2017*). Since each layer in DenseNet takes all previous layers as input, it tends to have more diverse features and richer patterns. In standard ConvNet, the input image is multiple convolutions to obtain high-level features for the classifier to use the most complex features. On the other hand, in DenseNet, the classifier tends to give smoother decision boundaries by using features of all complexity levels. This also explains why DenseNet outperforms other CNN architectures when training data is insufficient.

As shown in Fig. 4, DenseNet starts with a basic convolutional and a pooling layer. These form the basic building blocks of CNN architecture and are responsible for perceiving the features of the image.

The convolutional layer, which aims to learn the features of the data given as input, consists of a series of convolutional nuclei, with each neuron acting as a nucleus. The filter in the convolutional layer splits the image into smaller segments, which helps to extract feature motifs.

The pooling layer takes a small portion of the convolutional layer's output as input and samples it to produce a single output. Then comes a dense block, followed by a transition layer (made up of another convolution layer and another pooling layer), then another dense block, then a transition layer, another dense block, and then a classification layer. Each dense block has two convolution layers with cores of sizes $1 \times 1$ and $3 \times 3$. Dense blocks work with six, 12, 24, and 16 repetitions, respectively. After each dense block, the transition layer is used to control the complexity of the model as the number of channels increases. The transition decreases the number using the convolutional layer and reduces the complexity of the model by halving the height and width of the average pooling layer in step 2.

While the training process of DenseNet, Adam Optimizer was used. The epoch was set to 50. The mini-batch size was set to 32 and the learning rate was set to 0.001.

## One-shot learning and Siamese neural network

CNN achieves successful results in classification processes. However, the most basic problem of CNN and methods with similar infrastructure is the need for a large amount of

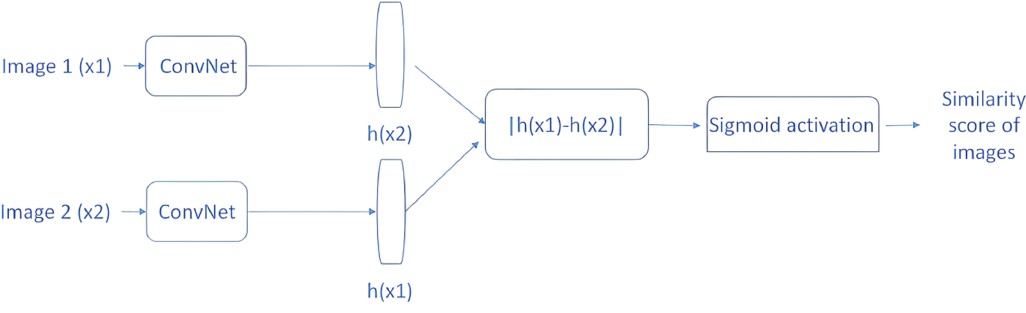

**Figure 5 SNN similarity calculation approach.**     

data. For some applications or problems, a large amount of data may not be available or obtainable.

The purpose of the OSL method is to train the model with only one or a few training examples. More specifically, the main idea of the SNN model is to train twin networks in a supervised method. Thus, the SNN consists of two identical sub-neural networks, each accepting an input. The outputs of the two subnets are forwarded to a middleware that calculates the distance between them.

The distance metric will be a high value if the two images in a pair come from two different classes, and a low value if they come from the same class. An example SNN model is shown in Fig. 5. The two CNN architectures shown are two copies of the same network, and they share the same parameters. First, two images ($\times 1$ and $\times 2$) are passed through ConvNet to generate a fixed-length feature vector for each (h($\times 1$) and h($\times 2$)).

Assuming that the model has been properly trained, the following assumptions can be made. If two input images belong to the same class, the feature vectors should also be similar; if the two input images belong to different classes, the feature vectors will also be different. Therefore, the absolute difference between the two feature vectors is calculated and the calculated value is passed through the dense layer with the sigmoid activation function for similarity score calculation. In light of the above information, the main idea behind the SNN can be explained.

## RESULT AND DISCUSSION

Once the pre-processing step was completed, the estimation of the building roof type was performed by feature extraction using an OSL approach on a small number of test images taken from the satellite.

During the training phase of the study, a random image was taken from any class. Then, a different image of the same class was taken; this image pair was labelled as '1'. In the next step, a random image belonging to a different class from the first image was taken and the pair of this new image and the first image was labelled as '0'. The goal was to provide a pair of images from the same class labelled as 1 and a pair of images from two different classes labelled as 0. Training results were recorded depending on the number of iterations given parametrically. The number of training iterations used in this study was determined as 100.

**Table 1 Classification examples.**

| Roof type | Compared roof type | Result |
|---|---|---|
| (a) Example 1 | | |
| Flat | Flat | 0.439 |
| Flat | Hip | −1.079 |
| Flat | Gable | 0.189 |
| (b) Example 2 | | |
| Gable | Gable | −4.694 |
| Gable | Hip | −0.119 |
| Gable | Flat | −2.892 |

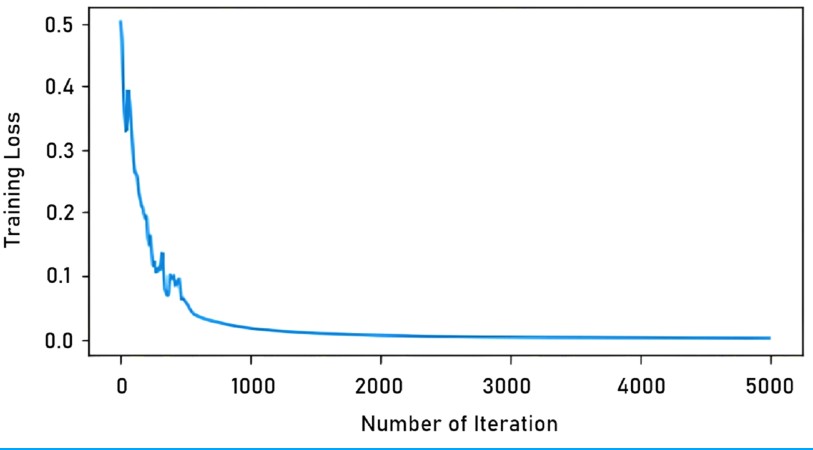

**Figure 6 Iteration-training loss graph.**

During the test phase, pairs were obtained in the same way as during the training phase. The number of images for the test was determined parametrically. Within the scope of this study, this value was chosen as 200. In other words, 200 image triads enter the testing phase. It was expected that the results produced by the image pairs of the same class would be the highest during the test process.

The first comparison (Example 1) in Table 1 is between flat classes. The highest result was 0.44, which shows the correct classification. But in Example 2, the Gable-Hip similarity was shown as higher than the Gable-Gable similarity. This indicates incorrect classification.

In our study, the total number of iterations in learning with a single image was determined as 5,000. In tests with more images, this number was 90,000. The reason for this difference is that learning with a single image is completed in a shorter time, as seen in Fig. 6. As can be seen in the graph, the model trained after 5,000 iterations the training loss is virtually unchanged.

The SNN model was first trained with a large amount of real roof images, then the data was reduced using one-shot logic and the study was repeated. Then, the same data sets were run with the CNN method and compared. The results in Table 2 were obtained by

**Table 2 Comparison of big data set (real roof images)—CNN *vs* SNN.**

| Amount of data | SNN model (accuracy) | CNN model (accuracy) |
| --- | --- | --- |
| 100 | 0.79 | 0.43 |
| 200 | 0.86 | 0.58 |
| 400 | 0.87 | 0.88 |
| 600 | 0.85 | 0.84 |
| 1,200 | 0.86 | 0.93 |
| 2,400 | 0.93 | 0.97 |

**Table 3 Simplified data set (real roof images) comparison—CNN *vs* SNN.**

| Amount of data | SNN model (accuracy) | CNN model (accuracy) |
| --- | --- | --- |
| 1 | 0.55 | 0 |
| 5 | 0.67 | 0 |
| 10 | 0.64 | 0 |
| 20 | 0.71 | 0 |
| 40 | 0.69 | 0 |
| 60 | 0.73 | 0.39 |

including 50% of the data set (*Alidoost & Arefi, 2018*) used in our study in the test and 100, 200, 400, 600, 1,200 and 2,400 data samples, respectively, in the training process. According to the results obtained, the SNN model gives better results with a small amount of data. On the other hand, the CNN model started to give better results with the amount of images exceeded 1,000.

Second, the results in Table 3 were obtained by reducing the number of images for each class. In this section, 1, 5, 10, 20, 40 and 60 real roof images were given to the model for training respectively, while a total of 60 real roof images randomly selected from 4,800 samples were used for the test. The average of the 25 results was obtained. The CNN and SNN methods were trained and tested with the same data set. Again, the SNN model proved more successful than the CNN model with fewer data. Meanwhile, the CNN model could not produce any results with up to 60 pieces of data.

Finally, we tested artificially produced roof types. As can be seen in Fig. 7, artificial roof images with different textures were produced in our study and 11 different test results were obtained with the same test set by giving training examples, one for each roof type. Each test result contained one flat, one gable and one hip roof sample. The most successful result was obtained with the image set in the $10^{th}$ test, where accuracy of 66.1% was achieved.

One-shot roof-type classification results using a single image and the SNN method are presented with the confusion matrix in Table 4. The accuracy, precision and recall performance metrics are tabulated in Table 5. The classification accuracy of the flat roof type was calculated as approximately 99%, with 47% for the Gable roof type and 55% for the Hip roof type. Thus, the average accuracy rate was 66.1%.

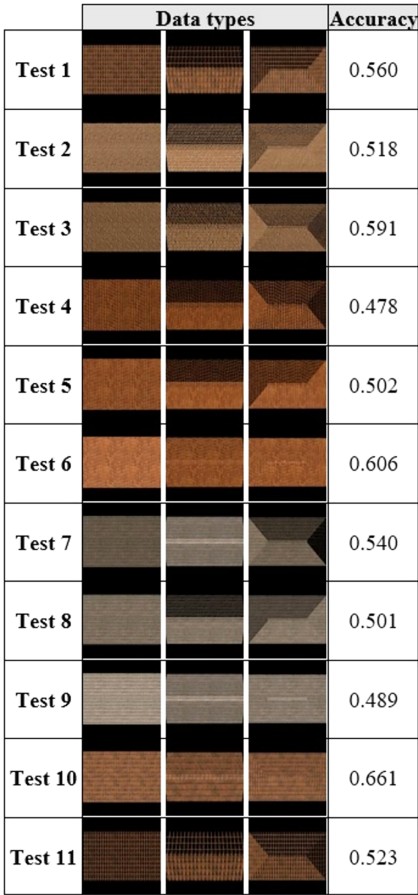

| | Data types | | | Accuracy |
|---|---|---|---|---|
| Test 1 | | | | 0.560 |
| Test 2 | | | | 0.518 |
| Test 3 | | | | 0.591 |
| Test 4 | | | | 0.478 |
| Test 5 | | | | 0.502 |
| Test 6 | | | | 0.606 |
| Test 7 | | | | 0.540 |
| Test 8 | | | | 0.501 |
| Test 9 | | | | 0.489 |
| Test 10 | | | | 0.661 |
| Test 11 | | | | 0.523 |

**Figure 7 Simplified data set (real roof image) comparison.**

**Table 4 Confusion matrix.**

| | Flat | Gable | Hip |
|---|---|---|---|
| Flat | 1,259 | 7 | 8 |
| Gable | 390 | 657 | 362 |
| Hip | 301 | 288 | 728 |

**Table 5 Precision and recall values by roof type.**

| | Accuracy | Precision | Recall |
|---|---|---|---|
| Flat | 0.99 | 0.65 | 0.99 |
| Gable | 0.47 | 0.70 | 0.47 |
| Hip | 0.55 | 0.66 | 0.55 |

The reason why the accuracy rate is very high in the flat roof type is that the test images do not contain much detail (as the roofs have a flat surface). Accordingly, as there are many details in the gable and hip roof images and the resolution of the satellite images is not very high, the performance rates remained lower with the other roof types.

In sum, this study showed that a model trained with just a single item of data could achieve satisfactory success in classification.

## CONCLUSION AND FUTURE WORK

This study aims to address the challenges of CNN-based methods that require a lot of training data to perform roof-type classification. Since the roof data set production is a difficult process, an artificial data set was produced. It is thought that this study will make a contribution to the literature in terms of producing artificial data to be used in the model training phase and training with a small amount of data.

Within the scope of the study, the SNN was first tested with large data sets, and then the data was reduced using the OSL approach. With a single training data, it gets approximate results with an accuracy of 99%, 47%, and 55% for flat roof, gable roof and hip roof types, respectively. Thus, the average accuracy rate was 66.1%. The same data sets were also tested with the CNN method and the results were compared. While results could not be obtained up to 60 training sets, SNN achieved 73% accuracy with 60 datasets, while CNN achieved 39% success. It has been observed that the CNN approach cannot get results using below a certain amount of data. It has been proven that the OSL approach can get satisfactory results even with just one data point.

It is predicted that these results using the one-shot method with artificial training data are promising and can be extrapolated onto a more general approach with different test data. Although the results of the model are satisfactory, it is thought that better results will be obtained with higher-resolution test data.

In future studies, it may be beneficial to diversify artificial data production by making use of different roof textures. The proposed method has been tested with images from only one area of the roof data set. Testing the method with different data sets will be one of our future studies.

### Funding
The authors received no funding for this work.

### Competing Interests
The authors declare that they have no competing interests.

### Author Contributions
- Naim Ölçer conceived and designed the experiments, performed the experiments, analyzed the data, performed the computation work, prepared figures and/or tables, authored or reviewed drafts of the article, and approved the final draft.

- Didem Ölçer conceived and designed the experiments, performed the experiments, prepared figures and/or tables, and approved the final draft.
- Emre Sümer conceived and designed the experiments, analyzed the data, authored or reviewed drafts of the article, and approved the final draft.

## Data Availability

The codes and data are available at GitHub: https://github.com/olcernaim/roof-type-classification; olcernaim. (2022). olcernaim/roof-type-classification: Roof type classification Release 1 (v1.0.0). Zenodo. https://doi.org/10.5281/zenodo.7436349.

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
