# Peer review of "Roof type classification with innovative machine learning approaches"

_PeerJ Computer Science, doi:10.7717/peerj-cs.1217_

## Round 0.1 · original submission · Major Revisions

Dear authors, we strongly advise you to respond to the comments, concerns, and questions raised by reviewers (especially the second reviewer), which are listed at the bottom of this letter, and then resubmit your paper once it has been revised.

Reviewer 1 ·

Basic reporting

1. The overall English quality is not satisfactory.
2. The research background is not enough, with only very limited references for roof type classification. The flaws of the existing roof type classification methods and the advantages of Siamese Neural Networks for similar problems are not summarized. The research gap and the research motivation for proposing a new machine learning based on Siamese Neural Network is not clear.
3. The figures are in a low quality.

Experimental design

1. The knowledge gap is not identified. The research design with two datasets and the usage of artificial data set instead of real-world data set is not justified.
2. The model details for both the DenseNet CNN model and the SNN model are not given, making the results less reliable.
3. There are more recent solutions for computer vision problems, and the choice of the SNN structure as the solution and the choice of DenseNet as the baseline are not so convincing.

Validity of the findings

1. The datasets are publicly available, but the replication is not possible since the source code is not provided.

·

Basic reporting

This paper is well written in general. The methodology is sound and the ideas are interesting to readers.

The authors studied the roof type image classification, with special interests in the application of Siamese network.

The authors proposed a method based on one shot training is investigated and large amount of training data is avoided. The literature review could be futher enhanced, more recent work within three years could be included and compared. Such as some new studies on Siamese network and their applications in retail image classificaion, remote sensing, etc.

Experimental design

Is the dataset used in this study public available, and where can the readers obtain such dataset used in the experiments?
DenseNet and other backbone architectures are discussed in this paper, I would suggest to more discussion on why the authors choose the specific network structure in the experiments.

The details of parameter settings for their experiments could be further provided for the interested readers to reproduce and compare the results.

Validity of the findings

The conlusion is supported by the experimental results.

---

## Round 0.2 · accepted · Accept

Thank you for the hard work in order to clearly and appropriately address all of the reviewers' comments.

Reviewer 1 ·

Basic reporting

Overall this version has improved a lot and no further comments.

Experimental design

no comment

Validity of the findings

no comment

Additional comments

no comment

·

Basic reporting

The revised verison has been improved. The revised paper is well written in general.

Experimental design

The authors have provided extended discussions and detials for the experiments. My previous concerns have been addressed.

Validity of the findings

The revised paper has been improved. Validity of the results are futher improved by discussions on the results.

Additional comments

My previous concerns have been addressed.